# Superstatistical analysis of sea surface currents in the Gulf of Trieste, measured by HF Radar, and its relation to wind regimes, using the maximum entropy principle

Sofia Flora[1,2], Laura Ursella[2], and Achim Wirth[3]

[1]University of Trieste, Dipartimento di Matematica e Geoscienze, Trieste, Italy
[2]Istituto Nazionale di Oceanografia e di Geofisica Sperimentale - OGS, Trieste, Italy
[3]Univ. Grenoble Alpes, CNRS, Grenoble INP, LEGI, 38000 Grenoble, France

**Correspondence:** Sofia Flora (sflora@ogs.it)

**Abstract.** Two years (2021-2022) of High Frequency Radar (HFR) sea surface current data in the Gulf of Trieste (Northern Adriatic Sea) are analysed. Two different time scales are extracted using superstatistical formalism: a relaxation time and a larger timescale over which the system is gaussian. We propose to obtain the ocean current Probability Density Function (PDF) combining: (i) a gaussian PDF for the fast fluctuations and (ii) a convolution of exponential PDFs for the slowly evolving variance of the gaussian function rather than for the thermodynamic $\beta = 1/\sigma^2$ in a system with few degrees of freedom as the latter has divergent moments. The Gaussian PDF reflects the entropy maximisation for real-valued variables with a given variance. On the other hand, if a positive variable, as is a variance, has a specified mean, the maximum entropy solution is an exponential PDF. In our case the system has two degrees of freedom and therefore the PDF of the variance is the convolution of two exponentials.

In the Gulf of Trieste there are three distinct main wind forcing regimes: Bora, Sirocco and low wind, leading to a succession of different sea current dynamics on different time scales. The universality class PDF successfully fits the observed data over the two observation years and also for each wind regime separately with a different variance of the variance PDF, which is the only free parameter in all the fits.

# 1  Introduction

Earth observing systems provide us with an ever increasing amount of data. This allows us today to move from considering low order moments of fluctuating observations, averages and variances, to their Probability Density Functions (PDFs), which give a complete characterization of the statistics and help exposing the underlying dynamics. These PDFs are mostly non-gaussian with fat tails, showing that extreme events are more frequent as compared to gaussian statistics. Furthermore, when the system is subject to slow external forcing, the PDF of the fast fluctuations, characterized by the signal decorrelation time, evolves on

a slower time scale. Examples of such dynamics are: fast weather statistics (days) under slowly evolving climat conditions (tens of years), or fast sea surface current fluctuations (hourly) during slowly varying synoptic weather conditions (few days), the problem considered here. Superstatistics, introduced by Beck and Cohen (2003), Beck (2004) and Beck et al. (2005b), is a formalism for considering dynamics on two separated time scales, as it considers a fast gaussian process whose variance evolves on a slower time scale. This leads to long term PDFs which are typically fat tailed.

Superstatistics is a method widely used today in various scientific fields: enviromental science (Weber et al., 2019; Schäfer et al., 2021), biology (Costa et al., 2022), statistical mechanics (dos Santos et al., 2022), quantum science (Okorie et al., 2023). Superstatistics is here applied to the Gulf of Trieste, which is used as a natural laboratory to explore the air-sea interaction. The Gulf of Trieste is a shallow semi-enclosed basin in the Northern Adriatic Sea (Mediterranean Sea; Fig. 1a): its maximum depth is approximately $25$ m and its shape is roughly a $20$ km $\times$ $20$ km square, opened on the western side. The Gulf of

Trieste sea surface base circulation is driven by the basin-wide Adriatic cyclonic (counter-clockwise) circulation and by the internal thermohaline properties of the basin (Cosoli et al., 2012; Querin et al., 2021), showing a mean surface water outflow (Bogunović and Malačič, 2008). But this base state is highly varied by two different forcings: the north freshwater input from the Isonzo/Soča River and the wind forcing (Querin et al., 2006; Cosoli et al., 2012; Cosoli et al., 2013; Querin et al., 2021). Moreover, the Gulf of Trieste together with the northernmost part of the Adriatic shelf are also sites of North Adriatic Dense

Water formation due to shelf convection (Jeffries and Lee, 2007; Pullen et al., 2007).

The main wind regimes consist in Bora and Sirocco wind events. The Bora is a east-north-easterly katabatic wind with gusts reaching up to $50$ m s$^{-1}$ and bringing cold and dry continental air from the north-east over the sea (Poulain and Raicich, 2001). The Bora intensity is highly affected by the land orography and it blows with particular strength over the Gulf of Trieste. Bora winds are most common during the cold season, but they can occur also during the summer. When the Bora blows, it forces

the surface water out of the Gulf and a replenishing bottom counter-current entering the Gulf causes an upwelling on its closed side (Querin et al., 2006; Malačič et al., 2001). Furthermore Bora events are very efficient in causing simultaneous cooling and evaporation by bringing cold and dry continental air in contact with the sea surface (Poulain and Raicich, 2001; Raicich et al., 2013). In contrast, the Sirocco is a southerly moist and warmer wind, channeled by the Adriatic coastal mountains that tends to occur year-round without a favored month or season (Poulain and Raicich, 2001). The relatively smooth Sirocco wind causes a

sea-level rise in the northern Adriatic which drives an intense southward return flow when wind forcing relaxes (Cosoli et al., 2012).

High Frequency Radar (HFR) is a powerful technology to measure horizontal surface currents and sea waves over a grid in a

wide area of the sea. The HFR technology is founded on the principle of Bragg scattering of the electromagnetic radiation over the rough conductive sea surface (Crombie, 1955). Currents are obtained from the Doppler shift of radio waves backscattered by surface gravity waves of half their electromagnetic wavelength. Each radar site allows to estimate the component of the current towards or away from the receiving antenna (radial current component), and thus two sites are needed to obtain total currents. The distance to the backscattered signal is determined by range-gating the returns. Depending on the methodology used to determine the incoming direction of the scattered signal (also called "bearing determination"), commercial HFR systems can be divided into two main types: Beam Forming and Direction Finding. Detailed description of radar technology and its uses can be found in Lorente et al. (2022).

This study aims to investigate the ocean currents in the Gulf of Trieste focusing on the fluctuations of the recorded signals. It is done adopting a superstatistical approach (Beck et al., 2005b). Superstatistics in fact is a method based on the development of statistics of statistics which is well suited to the case of the Gulf of Trieste, where different sea current dynamics on different time scales superpose. Many studies (e.g. Ghil et al., 2011 and references therein) use the maximum entropy principle as a tool in environmental science. Here, according to Jaynes (1957)'s point of view, the entropy maximisation will be applied to find the least biased PDFs.

The paper is organized as follows: in Sect. 2 the used data are presented, in Sect. 3 the adopted methodology is explained, in Sect. 4 the results are shown, finally the conclusions are in Sect. 5.

## 2  Data

Two classes of data sets are analysed: the sea surface horizontal HFR current data and the modelled wind data within the area of the Gulf of Trieste (Northern Adriatic Sea, Fig. 1). The available data sets cover a time range of almost two years: from January 1, 2021 to October 18, 2022.

### 2.1  The HFR current data

Sea surface currents consist of HFR combined current data coming from two beamforming WEllen RAdar stations operating in the Gulf of Trieste (Fig. 1b, red dots), the first located in Aurisina (Italy) and operated by the National Institute of Oceanography and Applied Geophysics (OGS) and the other in Piran (Slovenia), first operated by the National Institute of Biology (NIB) and later by the Slovenian Environmental Agency (ARSO). HFR works at the frequency of 24.5 MHz, with a space resolution of 1.5 km and a time resolution of 30 min. The data are freely available from the European HFR node at the following link: https://thredds.hfrnode.eu:8443/thredds/NRTcurrent/HFR-NAdr/HFR-NAdr_catalog.html.

The quality control standards from the EU high-frequency node (Corgnati et al., 2018) were applied to the data set. In addition, any remaining spikes were removed.

The grid points close to the HFR baseline have a major data coverage but show a more anisotropic behaviour than those far

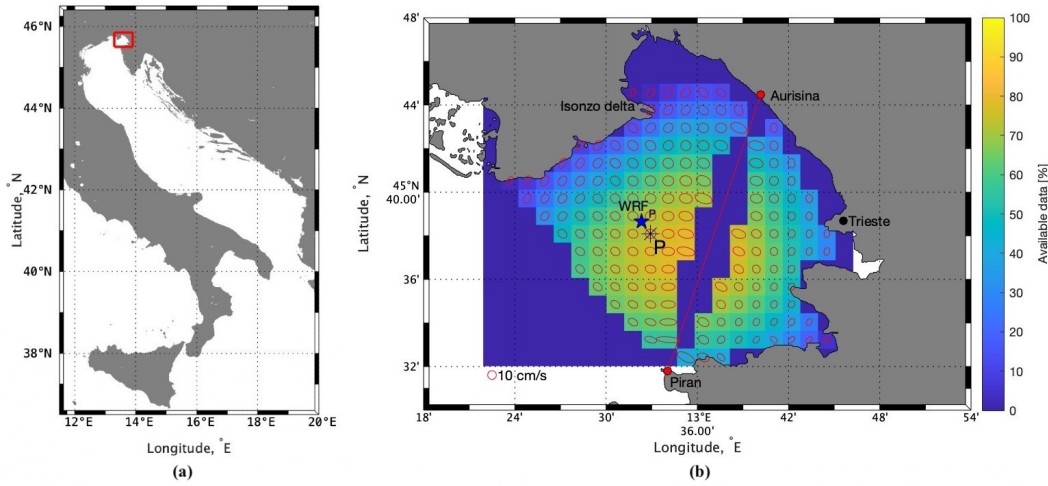

**Figure 1.** (a) Gulf of Trieste location (red rectangular) in the Adriatic Sea; (b) Gulf of Trieste zoom with percentage of available HFR data (in multiple colors) in the selected period and principal axes ellipses (in red) of the $\delta = 8 \times 30\text{min} = 4\text{h}$ velocity increments. The HFR baseline is shown with the red line between Aurisina and Piran. The HFR "P" grid point is shown with the black asterix and the closest WRF grid point is marked with the blue star and called "WRF$_\text{P}$". The selected HFR P grid point shows a high data coverage and is not close to the HFR baseline.

away (Fig. 1b). The P grid point was selected as the best candidate for the analysis of this paper, because it has a good data
80    coverage and the current increments are nearly isotropic.

## 2.2   Atmosphere data

The atmosphere data consist of the forecasted wind velocity field at 10 m above the surface in the Gulf of Trieste. These data are the output of the Weather Research and Forecasting (WRF) model (https://www.mmm.ucar.edu/models/wrf), version 4.2.1, performed daily by the Agenzia Regionale per la Protezione dell'Ambiente del Friuli Venezia Giulia (ARPA FVG) using
85    initial and boundary conditions from the National Oceanic and Atmospheric Administration Global Forecasting System (https: //www.ncei.noaa.gov/products/weather-climate-models/global-forecast). There are three calculation domains: the smallest one has the highest resolution, achieved with the two-way nesting technique, and provides the data used here (Goglio). The field has a spatial resolution of 2 km and a temporal resolution of 1 h.

Only the data from the closest WRF grid point to the P point are used for the following analysis. This point will be called
90    "WRF$_\text{P}$" and it is shown in Fig. 1b.

## 3    Method: Superstatistics

The analysis follows closely the framework introduced by Beck et al. (2005b). The reader familiar with it can skip the present section.

Consider a non-equilibrium system, characterized by a variable $v$ and driven by an intensive parameter $\sigma^2$ (the temperature in Boltzmann statistics). This parameter $\sigma^2$ is approximately constant on a time scale $T$ over which the system reaches a local equilibrium. The system fluctuates on a quicker decorrelation timescale $\tau \ll T$. Over long times, the shape of the probability distribution of $v$ is non-trivial and depends on the probability distribution of $\sigma^2$ and on the type of interaction between $\sigma^2$ and $v$. The simplest example is the case of a Brownian particle moving in one dimension with a velocity $v$, a variance $\sigma^2$ which is related to the temperature and where the local conditional distribution $p(v|\sigma^2)$ is gaussian (Maxwell-Boltzmann statistics in the canonical ensemble of statistical mechanics). According to the original point of view of Jaynes (1957), we interpret the local gaussianity of the velocity, given a certain temperature or variance, as the least biased PDF estimate, as it is based on entropy maximisation. Interpreting Shannon entropy as a measure of the lack of knowledge, it is possible to construct PDFs that maximise this lack of knowledge when some information is available. These obtained PDFs are the best estimates based on available information. We refer the reader to the Supplementary Material, Sect. S5 to see how a continuous real variable with a given mean and variance maximises the entropy if its PDF is gaussian.

Superstatistics considers dynamics where the temperature or the variance is slowly varying in time. When the decorrelation time is much faster than the time over which the variance evolves $\tau \ll T$, the dynamics results in a superposition of statistically different dynamics, so that the PDF of the velocity is:

$$p(v) = \int\limits_{0}^{\infty} f(\sigma^2) p(v|\sigma^2) d(\sigma^2) \tag{1}$$

where

$$p(v|\sigma^2) = \left( \frac{1}{2\pi\sigma^2} \right)^{1/2} e^{-\frac{v^2}{2\sigma^2}}, \tag{2}$$

is the local gaussian PDF of the velocity $v$ for a given $\sigma^2$ (the driving intensive parameter over $T$) and where $f(\sigma^2)$ is the PDF of $\sigma^2$.

Several studies (Beck, 2004; Beck et al., 2005a; Beck et al., 2005b) have shown that the thermodynamic beta $\beta = 1/\sigma^2$ PDF often falls into three main classes: (i) $\chi^2$; (ii) inverse-$\chi^2$; (iii) log-normal. The justification for the classes is often scant and other PDF shapes are also observed (Rizzo et al., 2004; Yalcin and Beck, 2013; Schäfer et al., 2021).

Starting from the longitudinal component $u(t)$ and the latitudinal component $v(t)$ of the sea surface currents in the P grid point in the Gulf of Trieste (Fig. 1b), we calculated the velocity increments $\delta u(t) = u(t+\delta) - u(t)$ and $\delta v(t) = v(t+\delta) - v(t)$, with the time increment $\delta = 2^j \times 30$ min, $j = 0, 1, 2, ..., 7$, since the focus is on fluctuations and not on averages. The first objective is to extract, from each of these time series, the two superstatistical timescales $\tau_\star$ and $T_\star$ (where $\star$ stands for $\delta u$ or $\delta v$) with $\tau_\star < T_\star$; secondly, to obtain $\sigma_\star^2(t)$ and look at its PDF $f(\sigma_\star^2)$.

First consider the timescale $\tau$: the relaxation time $\tau_\star$ is defined as the exponential decay time of the autocorrelation function of the time series $\delta u(t)$ (or $\delta v(t)$). For each $\delta$, it is possible to calculate the autocorrelation function $C_{\delta u}(\Delta t) = \langle \delta u(t' + \Delta t) \delta u(t') \rangle = \frac{\int_0^{t_{\max} - \Delta t} \delta u(t' + \Delta t) \delta u(t') dt'}{\int_0^{t_{\max} - \Delta t} \delta u(t') \delta u(t') dt'}$ (analogously for $\delta v$) and, from it, extract the time for which $C_\star(\tau_\star) = 1/e$. The time $\tau_\star$ is the decorrelation time of the signal.

Secondly, consider the timescale $T$: this timescale is the time for which the system is locally gaussian. It is possible to calculate it through the average kurtosis value of the velocity increment in function of the time:

$$\kappa_{\delta u}(\Delta t) = \frac{1}{t_{\max} - \Delta t} \int\limits_0^{t_{\max} - \Delta t} \frac{\langle (\delta u - \overline{\delta u})^4 \rangle_{t, \Delta t}}{\langle (\delta u - \overline{\delta u})^2 \rangle_{t, \Delta t}^2} dt \tag{3}$$

and analogously for $\delta v$, where $\langle \cdots \rangle_{t, \Delta t} = \frac{1}{\Delta t} \int_t^{t + \Delta t} \cdots dt$ is the internal $\Delta t$ time average and $\overline{\cdots}$ is the total time average. By definition, $T_\star$ is the time for which $\kappa_\star(T_\star) = 3$, which is the gaussian value.

A large time separation $\tau_\star \ll T_\star$ is fundamental, because it guarantees that a local gaussian equilibrium is reached. To determine the variance PDF $f(\sigma_\star^2)$ we also need a record length $T_{\text{obs}} \gg T_\star$.

As the reader can see from Eq. (2), the parameter $\sigma_\star^2$ is precisely the variance of the local gaussian distribution and, knowing $T_\star$, it is possible to calculate $\sigma_\star^2(t)$ as follows:

$$\sigma_{\delta u}^2(t) = \langle \delta u^2 \rangle_{t, T} - \langle \delta u \rangle_{t, T}^2 \tag{4}$$

and analogously for $\delta v$.

Superstatistics allows to uncover the physics of a non-equilibrium system through finding the PDF of a variable by separating the time scales and bringing out the evolution of the local equilibrium PDF.

## 4 Results and discussion

First we have identified and clustered the wind regimes blowing over the Gulf of Trieste in the analysed period. Since Bora and Sirocco are synoptic winds, the daily wind time series have been used (Fig. 2a). For each day, a wind regime is attributed: (i) low wind if the wind speed is $< 3$ m/s (green area); (ii) Mistral if the wind speed is $\geq 3$ m/s and its direction $\theta$ is $-67.5°\text{N} \leq \theta < 22.5°\text{N}$ (light blue area); (iii) Bora if the wind speed is $\geq 3$ m/s and its direction $\theta$ is $22.5°\text{N} \leq \theta < 112.5°\text{N}$ (blue area); (iv) Sirocco if the wind speed is $\geq 3$ m/s and its direction $\theta$ is $112.5°\text{N} \leq \theta \leq 206°\text{N}$ (red area).

It is interesting to see that Bora shows the highest speed peaks (Fig. 2a), reveiling to be the strongest wind forcing over the Gulf of Trieste, with a maximum daily wind speed of about $50$ km h$^{-1}$. About the wind regime occurence, as can be seen from the histogram in Fig. 2b, the Low wind regime is the most frequent (occurence of more than 60 %), while among strong wind regimes the Bora arises more often (occurence of more than 25 %). Sirocco develops less frequently (occurence of just below 10 %), while Mistral has an occurence of less than 1 % (it counts just five daily events in a period of almost two years, Fig. 2a) providing an insufficient statistics, so it will be ignored in the rest of the analysis.

Next we have considered the current data. Before following the $\tau_\star$ and $T_\star$ calculation procedure, the velocity increments PDFs

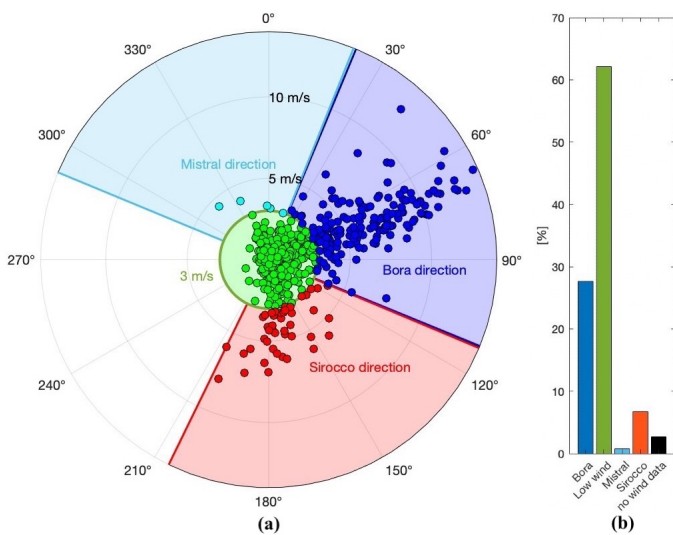

**Figure 2.** Classification of the wind regimes. (a): Daily wind (dots, the radial axes represents the speed and the azimuthal axes represents the direction) with threshold wind speed (3 m/s green line) and angle range (Mistral in light blue, Bora in blue and Sirocco in red) definition. Bora has the highest speed peaks; (b): Wind regime type normalized histogram. The main strong wind regimes are Bora and Sirocco.

were calculated: all these PDFs are non-gaussian and fat-tailed with kurtosis values greater than 3. The kurtosis value decreases with an increasing $\delta$, approaching the value 3, showing the fact that the larger the increment time, the more uncorrelated velocity
increments are and the closer the time series is to a gaussian process.

The velocity increments correlation functions were calculated together with the relaxation time $\tau_\star$ estimations, as explained in Sect. 3. The relaxation time $\tau_\star$ assumes values between approximately 15 min and 3 h, increasing with an increasing $\delta$. The velocity increments kurtosis values in function of the time were calculated together with the gaussianity timescale $T_\star$ estimations (Fig. 3), as explained in Sect. 3. The timescale $T_\star$ takes values longer than 10 h, increasing with $\delta$. This fact shows
that the larger the increment time, the more uncorrelated the time series is. We study the horizontal turbulent dynamics of the upper sea layer, evolving on time scales longer than a few hours, subject to the synoptic atmospheric forcing, with a typical time scale of several days. Choosing the velocity time increment $\delta = 8 \times 30 \min = 4$ h leads to $T_\star \simeq 2$ days, which is in agreement with the ocean surface-layer turbulent and the synoptic atmospheric time scales, respectively. In the following all the results will refer to this velocity time increment.

The resultant $\sigma_\star^2(t)$ time series was calculated, according to Eq. (4). It is possible to see that $\sigma_\star^2(t)$ varies more slowly than the velocity increment (Fig. 4). This fact is also apparent in the comparison between the velocity increments autocorrelation and the $\sigma_\star^2$ autocorrelation (not shown). It is possible to see that $\sigma_{\delta u}^2$ has higher peaks with respect to $\sigma_{\delta v}^2$. This means that $\delta u$ locally has a higher variance and thus a greater variability than $\delta v$. This is probably due to the geometry of the Gulf of Trieste whose western side is open. However this difference is not particularly pronounced, in fact it does not exceed one order of
magnitude. Moreover no clear seasonal cycle is evident.

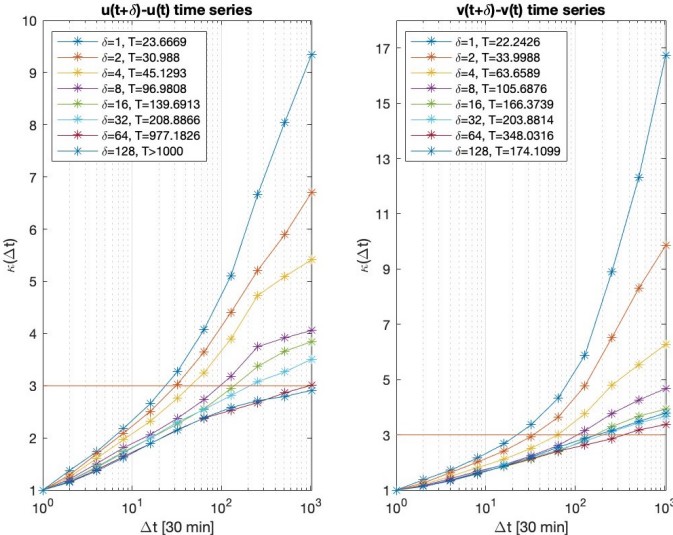

**Figure 3.** Velocity increments kurtosis value $\kappa_\star$ in function of time $\Delta t$ from the P grid point and $T_\star$ estimation for the $\delta u$ (left panel) and the $\delta v$ (right panel) velocity increments ($\delta$ and $T_\star$ in units of 30 min). The timescale $T_\star$ takes values always longer than 10 h, showing the time scale separation as $\tau_\star$ assumes values between 15 min and 3 h.

From our estimations on the observations (not shown) the thermodynamic betas $\beta_\star = 1/\sigma_\star^2$ do not fall into the $\chi^2$ and log-normal universality classes, but the fits suggest an Inverse-Gamma distribution (a universality class proposed by Beck et al., 2005b). In general, the Inverse-Gamma distribution with shape parameter $n$ has divergent $n^{th}$ and higher order moments, bringing limitations in fitting data for low $n$. As we will discuss below, we identify the shape parameter $n$ with the degrees of

freedom of the system, in our case equal to 2, leading to an analytical Inverse-Gamma distribution for $\beta_\star$ with only the first moment defined. For this reason we propose to consider $\sigma_\star^2 = \beta_\star^{-1}$ instead of $\beta_\star$. The PDF of the variance $f(\sigma_\star^2)$ is a Gamma distribution (Supplementary Material, Sect. S1) with fixed shape parameter equal to 2, where all positive moments converge:

$$f(\sigma_\star^2) = \Gamma_{2,\lambda_\star}(\sigma_\star^2) = \lambda_\star^2 \sigma_\star^2 e^{-\lambda_\star \sigma_\star^2}. \tag{5}$$

Our environmental system has two spatial degrees of freedom, so we can describe $\sigma_\star^2$ as the sum of two independent positive-

defined variables ($A_\star$ and $B_\star$). Again we apply entropy maximisation. Each of these independent variables is distributed to maximise entropy: since it is a positive variable, it is distributed as an exponential distribution, a particular case of the Gamma distribution (Supplementary Material, Sect. S5):

$$\sigma_\star^2 = \underbrace{A_\star + B_\star}_{A_\star \text{ and } B_\star \text{ are independent}} \quad ; \quad \text{where} \quad \begin{cases} A_\star \sim \Gamma_{1,\lambda_\star}(x) = \lambda_\star e^{-\lambda_\star x} \\ B_\star \sim \Gamma_{1,k_\star}(x) = k_\star e^{-k_\star x}. \end{cases} \tag{6}$$

This fact would give the following $\sigma_\star^2$ distribution (Supplementary Material, Sect. S2):

$$f(\sigma_\star^2) = \frac{\lambda_\star k_\star}{\lambda_\star - k_\star}(e^{-k_\star \sigma_\star^2} - e^{-\lambda_\star \sigma_\star^2}), \tag{7}$$

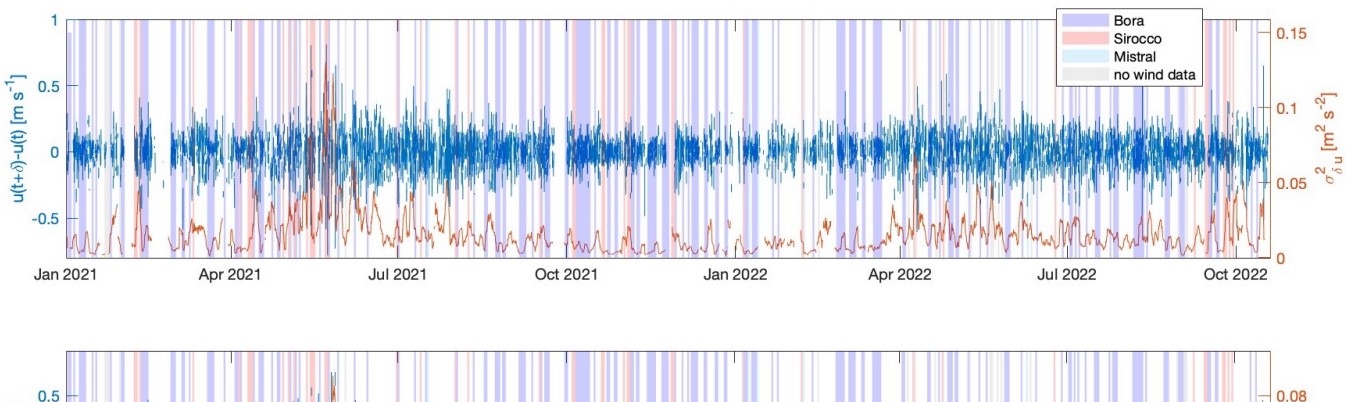

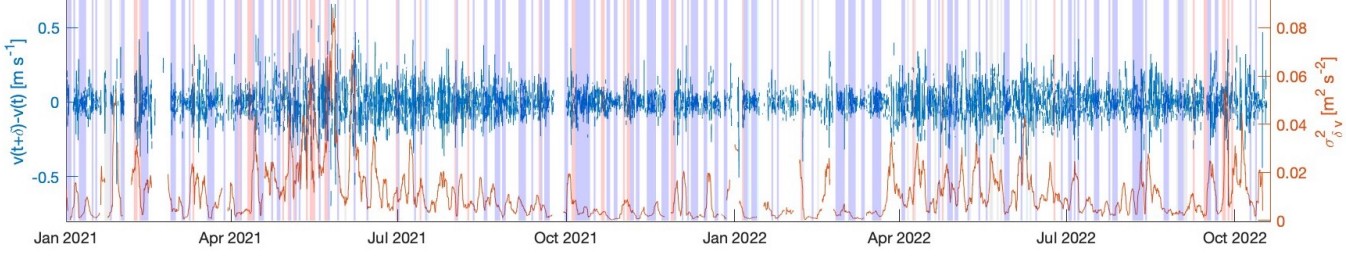

**Figure 4.** Velocity increments $\delta u(t)$, $\delta v(t)$ (in blue) and their respective $\sigma_\star^2(t)$ (in orange) time series with $\delta = 8 \times 30 \, \text{min} = 4 \, \text{h}$ from the P grid point and wind regimes periods from the $\text{WRF}_\text{P}$ grid point (coloured shades). The vertical scales are different in the two plots for better visualisation.

but, since the data are nearly isotropic (Fig. 1b), we can suppose the equality of the coefficient for both degrees of freedom $\lambda_\star = k_\star$ to obtain Eq. (5).

Using Eq. (1), Eq. (2) and Eq. (5), it is possible to calculate the PDF of the original signal (Supplementary Material, Sect. S3 and Sect. S4):

$$p(\delta u) = \int_0^\infty f(\sigma_{\delta u}^2) p(\delta u | \sigma_{\delta u}^2) d(\sigma_{\delta u}^2) = \frac{\sqrt{2\lambda_{\delta u}} e^{-\sqrt{2\lambda_{\delta u}}|\delta u|}(\sqrt{2\lambda_{\delta u}}|\delta u| + 1)}{4} \tag{8}$$

and analogously for $\delta v$. We emphasize that this PDF is obtained by maximasing entropy twice: for the gaussian distributed $p(\delta u | \sigma_{\delta u}^2)$ and for each degree of freedom of $f(\sigma_\star^2)$. We have fitted Eq. (8) on the observed $p(\delta u)$ and $p(\delta v)$ and their PDFs during the different wind regimes separately, obtaining for each curve the best coefficient $\lambda_\star$ (Fig. 5). As can be seen in Fig. 5, the analytical PDF expression is valid for the entire velocity increment data set as well as for the different wind regimes separately. From the expression in Eq. (8) it is possible to derive also the analytical expression of the velocity increment second order moments $s_\star^2$ (Supplementary Material, Sect. S3 and Sect. S4):

$$\left. \begin{array}{c} var(\delta u) \\ var(\delta v) \end{array} \right\} = s_\star^2 = \frac{2}{\lambda_\star} = \sqrt{2}\sigma_{\star\Gamma} = 2\sigma_{\star\exp} \tag{9}$$

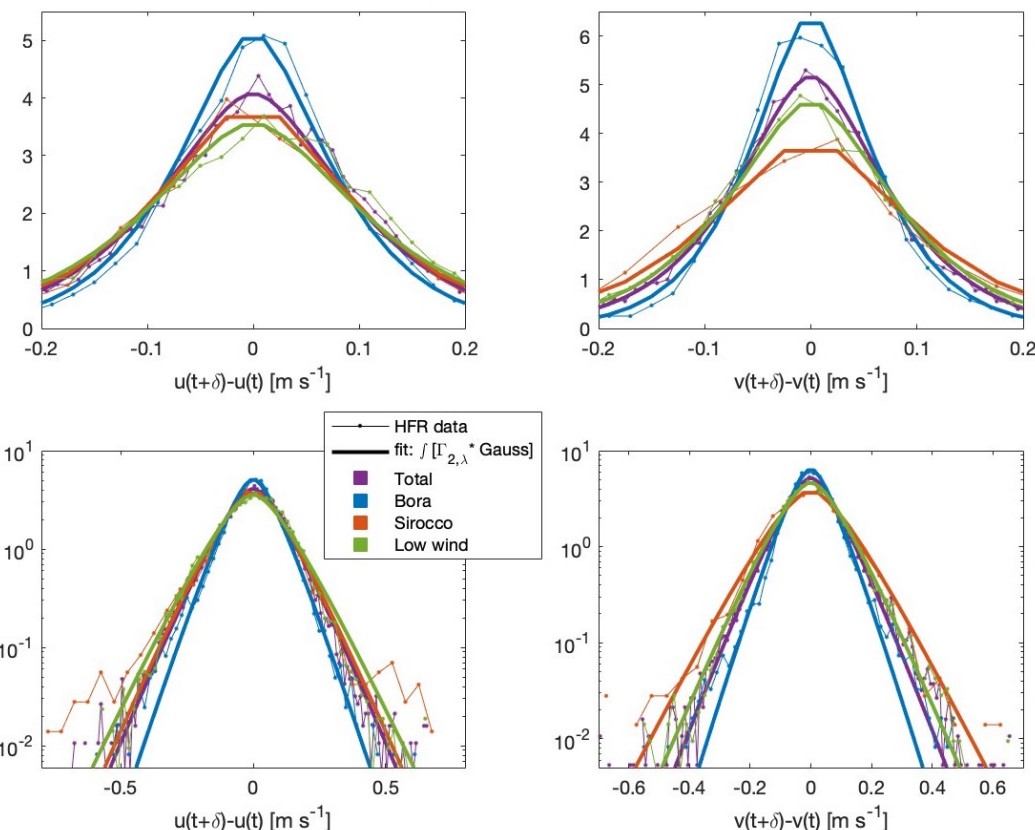

**Figure 5.** Velocity increments PDFs ($\delta u$ on left panels, $\delta v$ on right panels) from the P grid point and $\delta = 8 \times 30\text{min}$ with the best fit from Eq. (8). The PDFs are reported in a lin-lin plot (top panels), to enhance the peaks, and in a log-log plot (bottom panels), to enhance the tails. The thin spotted line represents the HFR data, the thick line represents the fit, while the different colours are used to discriminate between different wind regimes.

that are analytically related to the $\Gamma_{2,\lambda_\star}(\sigma_\star^2)$ standard deviation ($\sigma_{\star\Gamma}$) and to the exponential distribution standard deviation ($\sigma_{\star\exp}$). The velocity increment variance $s_\star^2$ is twice the exponential distribution standard deviation $\sigma_{\star\exp}$, as there are two spatial degrees of freedom. Equation (8) can then be expressed as:

$$p(\delta u) = \frac{e^{-2|\delta u|/s_{\delta u}}\left(\frac{2|\delta u|}{s_{\delta u}} + 1\right)}{2 s_{\delta u}}. \tag{10}$$

The $s_\star$ standard deviations numerical values from the best fit coefficients $\lambda_\star$ are reported in Table 1.

Using the best fit coefficients $\lambda_\star$ just derived, the solution of Eq. (5) is plotted and compared to the observed $\sigma_\star^2$ PDFs in Fig. 6.

| [cm/s] | Total | Bora | Sirocco | Low wind |
|---|---|---|---|---|
| $s_{\delta u}$ | 12.265 | 9.7675 | 12.8214 | 14.0345 |
| $s_{\delta v}$ | 9.6696 | 7.7663 | 12.9276 | 10.7316 |

**Table 1.** Best fits standard deviations from Eq. (9). Bora, the strongest wind forcing, leads to the lowest fluctuations of the velocity increments.

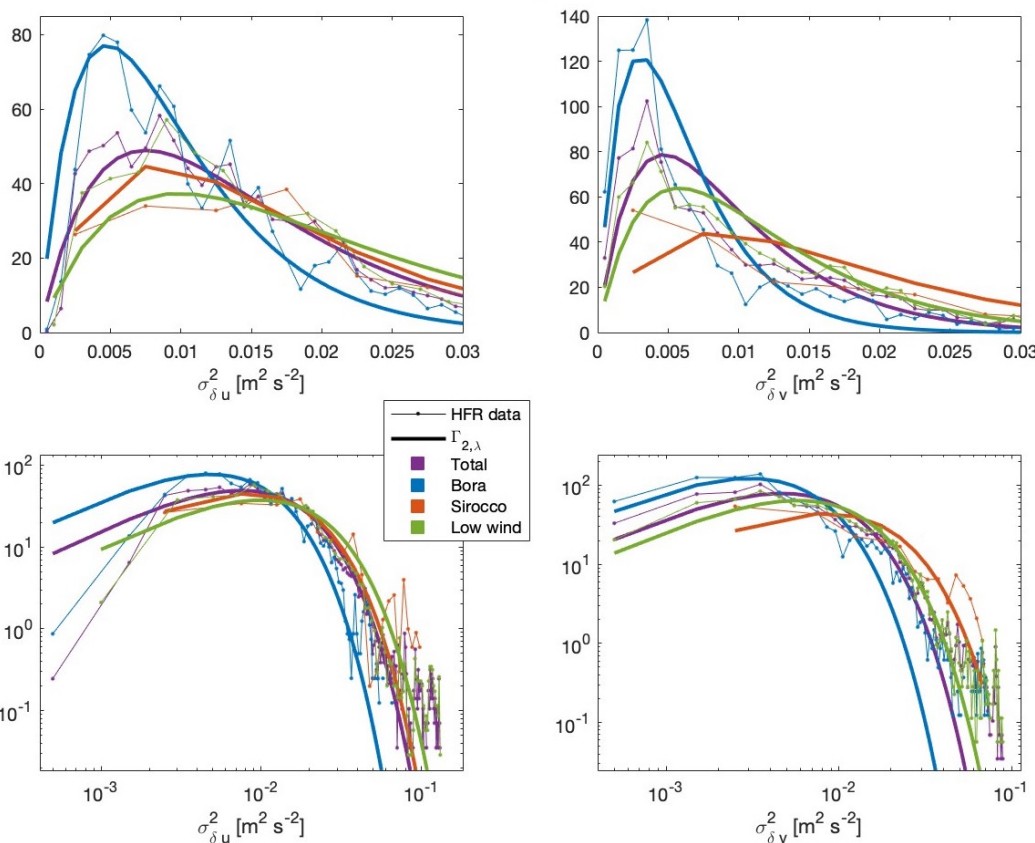

**Figure 6.** PDFs of $\sigma_{\delta u}^2$ (left panels) and $\sigma_{\delta v}^2$ (right panels) with $\delta = 8 \times 30$ min at P HFR grid point: Total $f(\sigma_\star^2)$ and clustered $f(\sigma_\star^2 | e)$ by the wind regime $e$. For the clustering operation, the $\sigma_\star^2$ time series were shifted by $T_\star/2$ and then the values were collected as a function of the daily wind regime. The PDFs are reported in a lin-lin plot (top panels) and in a log-log plot (bottom panels). The thin spotted line represents the HFR data, the thick line represents the analytical distribution, while the different colours are used to discriminate between different wind regimes.

## 5   Conclusions

We have observed that the sea surface currents velocity increments show fat tailes in their distributions. Superstatistics is a powerful tool for determining the PDF. In practice, it relies on the time scale separation between the fast fluctuations (for

which the Maxwell-Boltzmann statistics applies) and their slow driver (the temperature in statistical thermodynamics), which makes the local PDF to evolve with time. From the velocity increments time series we have extracted the two different time scales: the relaxation time $\tau_\star \simeq 1$ h 50 min and the larger $T_\star \simeq 2$ days, that is the gaussianity time scale. According to the

original point of view of Jaynes (1957), the local gaussianity of the velocity increments is the least biased PDF estimate on the information available as it reflects the maximisation of entropy. From the $T_\star$ time scale we have extracted the local gaussian variance time series $\sigma_\star^2(t)$ (the "temperature"). The maximum entropy PDF for a positive variable, as a variance, is exponential. In our case of two degrees of freedom we obtained the convolution of two identical exponentials that is a Gamma $\Gamma_{2,\lambda_\star}$ distribution function. Note that Beck and Cohen (2003) interprets the degrees of freedom as the number of the elements in

the sum of squared identically distributed gaussian variables, which give a gamma distributed variable. The degrees of freedom in their interpretation correspond to twice our value. The $\sigma_\star^2$ Gamma distribution corresponds to the Inverse-Gamma $\beta_\star$ class described by Beck et al. (2005b). Nevertheless, this distribution, representing a system with $n$ degrees of freedom, presents divergent $m \geq n$ order moments, bringing limitations in fitting data for low $n$, as in the present case. This problem completely disappears when the $\sigma_\star^2 = \beta_\star^{-1}$ is considered.

From the PDF of the variance we obtained the analytical distribution of the velocity increments. Fitting the distribution to our data we observe a good agreement, even if superstatistics works in the limit of infinite time-scale separations $\tau_\star \ll T_\star$ (a factor of few tens in our case) and $T_\star \ll T_{\mathrm{obs}}$ (a factor of few hundreds in our case). When the analysis is performed for the different wind regimes separately, the same analytical laws of the PDFs are obtained with different variance (Table 1), which is the only free parameter in the fit. The good agreement also for the different wind regimes is surprising as the length of the wind regimes

is comparable to $T_\star$ and for Sirocco winds we have only a few tens of independent events to fit the PDF of the variance. The good fits for the different wind regimes with the same analytic form points towards a universal behaviour. We showed that superstatistics is more than a tool for fitting fat-tailed PDFs: it gives the characteristic time scales and when combined with the maximum entropy principle it indicates the number of the degrees of freedom.

The velocity-increments second order moments differ between the forcing regimes (Bora, Sirocco and low wind), with the

strongest wind forcing (Bora) leading to the lowest fluctuations of the velocity increments. This might be a consequence of the sea surface current tendency to settle on a mean outflow (from the Gulf) during Bora or might indicate that an increased wave activity and turbulent exchange in the vertical of the horizontal momentum, operating at even shorter time scales than $\delta$, lead to a damping of the sea surface current fluctuations.

Parameterizations of turbulent motions generally rely on a velocity scale and a length scale, averaged over a large time scale.

The availability of a considerable amount of data allows us here to go further by determining the PDF of the velocity fluctuations at different time scales. More precisely, the PDF of the fast velocity fluctuations is gaussian and the PDF of the velocity variance at time scales larger than $T$ is $\Gamma_{2,\lambda}$. As already mentioned, for the Gulf of Trieste $T \simeq 2$ days for velocity increments over few hours (Fig. 3). The analytic form of the combined PDF of the velocity increments (Eq. (8)) can be used to develop parameterizations of the dynamics in the ocean surface layer at time scales faster than a few days and can discriminate between

existing parameterizations. Our findings show that parameterizations of surface current fluctuations at time scales shorter than $T \sim 2$ days can rely on gaussian fluctuations. For longer time scales the dynamics is non-gaussian and fat tailed, meaning that

extreme events are more ponderous and a parameterization has to account for it. The findings currently guide us in constructing stochastic differential equations that will lead to the same PDFs obtained here from a superstatistical analysis of the data. When the stochastic differential equation is known, it is mathematically straightforward to derive parameterizations of the turbulent dynamics.

*Data availability.* The HFR sea surface current data of the Gulf of Trieste are freely available from the European HFR node at the following link: https://thredds.hfrnode.eu:8443/thredds/NRTcurrent/HFR-NAdr/HFR-NAdr_catalog.html.
The WRF forecasted wind field is obtainable upon request from ARPA FVG (https://www.arpa.fvg.it)

*Author contributions.* SF performed the major part of the research and the writing of the manuscript. LU and AW contributed to the research and the writing.

*Competing interests.* The contact author has declared that neither of the authors has any competing interests.

*Acknowledgements.* The forecasts, analyses and related services are based on data and products of the Regional Center for Environmental Modelling (CRMA) which is a sector of the Friuli Venezia Giulia Environmental Agency (ARPA FVG) ITALY (https://www.arpa.fvg.it). Piran HFR station is operated by Slovenian Environment Agency since July 2022. Between 2015 and July 2022 it was operated by the National Institute of Biology.
This work was funded by "Initiatives de recherche à Grenoble Alpes (IRGA) 2022" through the project FASIOM.
SF acknowledges the hospitality of the LEGI laboratory (Grenoble, France)

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
