# Peer review of "Superstatistical analysis of sea surface currents in the Gulf of Trieste, measured by HF Radar, and its relation to wind regimes, using the maximum entropy principle"

_Nonlinear Processes in Geophysics, 2023_

## Author Comment (AC1)

Answers to both reviewers:

Dear Reviewers,

We are grateful to both reviewers for their corrections and comments as they have increased the quality of the paper.
Please find our detailed answers (written in blue) and corrections to both reviewers comments (reproduced in black) below. The corrections performed to the manuscript are given in red and an updated version of the manuscript with the corrections highlighted in red is provided.
Sincerely,
Sofia Flora, Laura Ursella and Achim Wirth

Reviewer 1:

The authors apply the formalism of superstatistics to describe the probability densities of velocity fluctuations in ocean currents relevant to forecasting in a region of Italy (Gulf of Trieste). The manuscript is interesting for two main reasons. First, it provides new tools for the statistical description of weather-relevant data that may not be that well-known in that community. Second, it reveals a new kind of superstatistical model, although somewhat related to the inverse-gamma (inverse chi-squared) superstatistics which is one of the main models originally proposed by Beck and Cohen. I would suggest improving the manuscript by addressing the following points:

1) Present in more detail the new model proposed in the work, namely Eq. 8 (equivalently, Eq. 10) and its relation to the inverse chi-squared superstatistics, taking into account that $1/\sigma^2$ plays the role of the inverse temperature beta when superstatistics is applied as an extension of statistical mechanics.
The referee is right, we are grateful to the reviewer for pointing out this important and not obvious point. If we express $\sigma^2$ distribution in terms of $\beta$ we obtain the inverse-gamma distribution with degrees of freedom $n = 2$. This PDF, however, has divergent second and higher order moments. This is a limitation to fitting the data for $n \leq 2$. As we used the variance for the fit, we missed this PDF. This problem completely disappears when the "temperature" $\sigma^2 = 1/\beta$ is considered, as we did. For lower degrees of freedom the consideration of $\sigma^2$ instead of $\beta$ is the better choice. In the paper we propose this new approach.
We modified in the Abstract:
"We propose to obtain the ocean current Probability Density Function (PDF) combining: (i) a gaussian PDF for the fast fluctuations and (ii) a convolution of exponential PDFs for the slowly evolving variance of the gaussian function rather than for the thermodynamic $\beta = 1/\sigma^2$ in a system with few degrees of freedom as the latter has divergent moments."
We modified in Sect. 4:
"From our estimations on the observations (not shown) the thermodynamic betas $\beta_\star = 1/\sigma_\star^2$ do not fall into the $\chi^2$ and log-normal universality classes, but the fits suggest an Inverse-Gamma distribution (a universality class proposed by Beck et al., 2005b). In general, the Inverse-Gamma distribution with shape parameter $n$ has divergent $n^{th}$ and higher order moments, bringing limitations in fitting data for low $n$. As we will discuss below, we identify the shape parameter $n$ with the degrees of freedom of the system, in our case equal to 2, leading to an analytical Inverse-Gamma distribution for $\beta_\star$ with only the first moment defined. For this reason we propose to consider $\sigma_\star^2 = \beta_\star^{-1}$ instead of $\beta_\star$. The PDF of the variance $f(\sigma_\star^2)$ is a Gamma distribution (Supplementary Material, Sect. S1) with fixed shape parameter equal to 2, where all positive moments converge:"
We added in Sect. 5:
The $\sigma_\star^2$ Gamma distribution corresponds to the Inverse-Gamma $\beta_\star$ class described by Beck et al. (2005)b. Nevertheless, this distribution, representing a system with $n$ degrees of freedom, presents divergent $m \geq n$ order moments, bringing limitations in fitting data for low $n$, as in the present case. This problem completely disappears when the $\sigma_\star^2 = \beta_\star^{-1}$ is considered.
Also, it would benefit the manuscript to give a larger context to the uses of superstatistics, as something beyond the mere description of two different time scales.
We added in Sect. 3:
"Superstatistics allows to uncover the physics of a non-equilibrium system through finding the PDF of a variable by separating the time scales and bringing out the evolution of the local equilibrium PDF."

We added in Sect. 5:

"Superstatistics is a powerful tool for determining the PDF. In practice, it relies on the time scale separation between the fast fluctuations (for which the Maxwell-Boltzmann statistics applies) and their slow driver (the temperature in statistical thermodynamics), which makes the local PDF to evolve with time. From the velocity increments time series we have extracted the two different time scales..."

"From the $T_\star$ time scale we have extracted the local gaussian variance time series $\sigma_\star^2(t)$ (the "temperature")."

2) Expand on the maximum entropy (MaxEnt) principle, a topic that may not be that well-known to the target audience, perhaps by providing the original point of view by E. T. Jaynes in that MaxEnt allows for the construction of minimally biased models given some piece of information.

We added in Sect. 3:

"According to the original point of view of Jaynes (1957), we interpret the local gaussianity of the velocity, given a certain temperature or variance, as the least biased PDF estimate, as it is based on entropy maximisation. Interpreting Shannon entropy as a measure of the lack of knowledge, it is possible to construct PDFs that maximise this lack of knowledge when some information is available. These obtained PDFs are the best estimates based on available information. We refer the reader to the Supplementary Material, Sect. S5 to see how a continuous real variable with a given mean and variance maximises the entropy if its PDF is gaussian."

We added in Sect. 4:

"Again we apply entropy maximisation. Each of these independent variables is distributed to maximise entropy: since it is a positive variable, it is distributed as an exponential distribution, a particular case of the Gamma distribution (Supplementary Material, Sect. S5)"

We added in Sect. 5:

"According to the original point of view of Jaynes (1957), the local gaussianity of the velocity increments is the least biased PDF estimate on the information available as it reflects the maximisation of entropy."

Finally, of course, we added Sect. S5 to the Supplementary Material document.

3) Fix some problems in the figures where NaN appears in the labels (Figs. 3 and 4).

The figures have now proper labels.

Thank you.

Reviewer 2:

In the article the authors claim that "A new analytical universality class of Probability Density Functions (PDFs) is proposed for ocean current data combining a gaussian PDF for the fast fluctuations and a convolution of exponential PDFs for the slowly evolving variance of the gaussian". Experimental data and simulated data (Weather Research and Forecasting (WRF) model) are used to Justify the conclusions. The topic seems to be interesting for the geophysics community and useful for NPG readers. To be more useful to the readers, I suggest the following point to be considered by the authors.

Please see also our answer to Reviwer 1.

1- Please provide more details in the abstract sentence: "The Gaussian PDF has maximum entropy for real-valued variables with a given variance."

The sentence is now changed to:

"...combining: (i) a gaussian PDF for the fast fluctuations and (ii) a convolution of exponential PDFs for the slowly evolving variance of the gaussian function [...]. The Gaussian PDF reflects the entropy maximisation for real-valued variables with a given variance."

2- Still in the abstract, some explanation are given in next sentences but I belive it is not completely consistent "If a positive variable, as is a variance, has a specified mean, the maximum entropy solution is an exponential PDF. Here it is the sum of

two exponentials, reflecting the two spatial degrees of freedom."

95 The sentence is now changed to:

"On the other hand, if a positive variable, as is a variance, has a specified mean, the maximum entropy solution is an exponential PDF. In our case the system has two degrees of freedom and therefore the PDF of the variance is the convolution of two exponentials."

100 3- Bibliographical references presented in the introduction are outdated.

We added in Sect. 1:

"Superstatistics is a method widely used today in various scientific fields: enviromental science (Weber et al., 2019; Schäfer et al., 2021), biology (Costa et al., 2022), statistical mechanics (dos Santos et al., 2022), quantum science (Okorie et al., 2023)."

Many works on methods based on entropy computation have been published and can be explored for a better explanation of
105 the results.

We added in Sect. 1:

"Many studies (e.g. Ghil et al., 2011 and references therein) use the maximum entropy principle as a tool in environmental science. Here, according to Jaynes (1957)'s point of view, the entropy maximisation will be applied to find the least biased PDFs."

110 4- Figure captions are not informative, only descriptive. Please describe the main points of each figure so that the information is self-consistent.

The captions are now changed to:

"Figure 1. (a) Gulf of Trieste location (red rectangular) in the Adriatic Sea; (b) Gulf of Trieste zoom with percentage of available HFR data (in multiple colors) in the selected period and principal axes ellipses (in red) of the $\delta = 8 \times 30\text{min} = 4\text{h}$ velocity
115 increments. The HFR baseline is shown with the red line between Aurisina and Piran. The HFR "P" grid point is shown with the black asterix and the closest WRF grid point is marked with the blue star and called "WRF$_\text{P}$". The selected HFR P grid point shows a high data coverage and is not close to the HFR baseline."

"Figure 2. Classification of the wind regimes. (a): Daily wind (dots, the radial axes represents the speed and the azimuthal axes represents the direction) with threshold wind speed (3 m/s green line) and angle range (Mistral in light blue, Bora in blue and
120 Sirocco in red) definition. Bora has the highest speed peaks; (b): Wind regime type normalized histogram. The main strong wind regimes are Bora and Sirocco."

"Figure 3. Velocity increments kurtosis value $\kappa_\star$ in function of time $\Delta t$ from the P grid point and $T_\star$ estimation for the $\delta u$ (left panel) and the $\delta v$ (right panel) velocity increments ($\delta$ and $T_\star$ in units of 30 min). The timescale $T_\star$ takes values always longer than 10 h, showing the time scale separation as $\tau_\star$ assumes values between 15 min and 3 h."

125 "Figure 4. Velocity increments $\delta u(t)$, $\delta v(t)$ (in blue) and their respective $\sigma_\star^2(t)$ (in orange) time series with $\delta = 8 \times 30 \text{ min} = 4$ h from the P grid point and wind regimes periods from the WRF$_\text{P}$ grid point (coloured shades) . The vertical scales are different in the two plots for better visualisation."

"Figure 5. Velocity increments PDFs ($\delta u$ on left panels, $\delta v$ on right panels) from the P grid point and $\delta = 8 \times 30\text{min}$ with the best fit from Eq. (8). The PDFs are reported in a lin-lin plot (top panels), to enhance the peaks, and in a log-log plot (bottom
130 panels), to enhance the tails. The thin spotted line represents the HFR data, the thick line represents the fit, while the different colours are used to discriminate between different wind regimes."

"Table 1. Best fits standard deviations from Eq. (9). Bora, the strongest wind forcing, leads to the lowest fluctuations of the velocity increments."

"Figure 6. PDFs of $\sigma_{\delta u}^2$ (left panels) and $\sigma_{\delta v}^2$ (right panels) with $\delta = 8 \times 30$ min at P HFR grid point: Total $f(\sigma_\star^2)$ and clustered
135 $f(\sigma_\star^2|e)$ by the wind regime $e$. For the clustering operation, the $\sigma_\star^2$ time series were shifted by $T_\star/2$ and then the values were collected as a function of the daily wind regime. The PDFs are reported in a lin-lin plot (top panels) and in a log-log plot (bottom panels). The thin spotted line represents the HFR data, the thick line represents the analytical distribution, while the different colours are used to discriminate between different wind regimes."

140 5- Figure 2 (b), in particular, is not explored at any time in the text.

We thank the reviewer, but we note that the reference to Fig. 2b is in line 132 of the first version of the manuscript. We can improve its discussion as suggested in the next comment.

6- The authors mention that "It is interesting to see that, if the wind is strong enough, the main wind regimes are just Bora and
145     Sirocco (Fig. 2b)" But it is not clear how this information can be observed in fig. 2(b).

The mentioned and the following sentences are now clarified:

"It is interesting to see that Bora shows the highest speed peaks (Fig. 2a), reveiling to be the strongest wind forcing over the Gulf of Trieste, with a maximum daily wind speed of about $50\,\mathrm{km\,h^{-1}}$. About the wind regime occurence, as can be seen from the histogram in Fig. 2b, the Low wind regime is the most frequent (occurence of more than 60 %), while among strong wind
150     regimes the Bora arises more often (occurence of more than 25 %). Sirocco develops less frequently (occurence of just below 10 %), while Mistral has an occurence of less than 1 % (it counts just five daily events in a period of almost two years, Fig. 2a) providing an insufficient statistics, so it will be ignored in the rest of the analysis."

We also added detailes in the figures, see above.

155     7- Section 4 is heavily descriptive also. I believe that more discussions and details about the entropy maximization method can be presented. Even the concept of super statistics may not be fully understood by the geophysics community.

We hope that the implementations in the text shown in the answers to the second part of comment 1) and comment 2) of Reviewer 1 are sufficient (in particular the addition of Sect. S5 in the Supplementary Material document).

160     8- Some values do not appear in Figure 3 (delta case 128).

The figure has now proper labels.

9- Figure 4 is not very informative. Perhaps it should be discussed further.

We added in Sect. 4:
165     "Moreover no clear seasonal cycle is evident."

In general, the article presents an analysis method that may be of interest to the geophysics community. After the suggested corrections/additions of information, I believe that the article may be accepted for publication.

Thank you.

**References**

Beck, C., Cohen, E. G., and Swinney, H. L.: From time series to superstatistics, Physical Review E, 72, 056 133, 2005.

Costa, M. O., Silva, R., and Anselmo, D. H. A. L.: Superstatistical and DNA sequence coding of the human genome, Phys. Rev. E, 106, 064 407, https://doi.org/10.1103/PhysRevE.106.064407, 2022.

dos Santos, M., Menon, L., and Cius, D.: Superstatistical approach of the anomalous exponent for scaled Brownian motion, Chaos, Solitons & Fractals, 164, 112 740, https://doi.org/https://doi.org/10.1016/j.chaos.2022.112740, 2022.

Ghil, M., Yiou, P., Hallegatte, S., Malamud, B., Naveau, P., Soloviev, A., Friederichs, P., Keilis-Borok, V., Kondrashov, D., Kossobokov, V., et al.: Extreme events: dynamics, statistics and prediction, Nonlinear Processes in Geophysics, 18, 295–350, 2011.

Jaynes, E. T.: Information theory and statistical mechanics, Physical review, 106, 620, 1957.

Okorie, U. S., Ikot, A. N., Okon, I. B., Obagboye, L. F., Horchani, R., Abdullah, H. Y., Qadir, K. W., and Abdel-Aty, A.-H.: Exact solutions of $\kappa$-dependent Schrödinger equation with quantum pseudo-harmonic oscillator and its applications for the thermodynamic properties in normal and superstatistics, Scientific Reports, 13, 2108, https://doi.org/10.1038/s41598-023-28973-7, 2023.

Schäfer, B., Heppell, C. M., Rhys, H., and Beck, C.: Fluctuations of water quality time series in rivers follow superstatistics, Iscience, 24, 102 881, 2021.

Weber, J., Reyers, M., Beck, C., Timme, M., Pinto, J. G., Witthaut, D., and Schäfer, B.: Wind power persistence characterized by superstatistics, Scientific reports, 9, 1–15, 2019.